# Working from Home and Emotional Well-Being during Major Daily Activities

**DOI:** 10.3390/ijerph20043616

**Published:** 2023-02-17

**Authors:** Brandon J. Restrepo, Eliana Zeballos

**Affiliations:** Economic Research Service, U.S. Department of Agriculture, Washington, DC 20250, USA

**Keywords:** work from home, work away from home, well-being, American Time Use Survey, Well-Being Module, D13, I12, I14, J22

## Abstract

The effect of WFH (working from home) on the quality of life of U.S. workers is not well understood. We analyze the association between WFH and overall emotional well-being during major daily activities. Using data from the 2021 Well-Being Module of the American Time Use Survey, we conduct a principal component analysis to construct a measure of overall emotional well-being and jointly estimate the association between WFH and overall emotional well-being scores in a seemingly unrelated regression framework. Our results show that compared to workers who worked outside the home, those who WFH had higher emotional well-being scores while working and eating away from home. However, no statistically significant differences were found for home-based daily activities such as relaxing, leisure, food preparation, and eating at home. These findings inform how WFH may shape the quality of a life day.

## 1. Introduction

The share of the U.S. workforce holding a job that is compatible with remote work has exploded in recent years, especially after the onset of the COVID-19 pandemic. Indeed, telework participation surged for many demographic subgroups from February to May 2020 [1], with 1 in 2 U.S. workers reporting in May 2020 that they were WFH (working from home) [2]. According to Barrero et al. (2021) [3], while fewer than 5 percent of paid full workdays were WFH just before the pandemic (March 2020), the share jumped to over 60 percent shortly after the pandemic began (May 2020), and has stabilized at about 30 percent since 2021.

Although a much broader swath of U.S. workers are WFH than before the pandemic, the effects on their quality of life are not yet fully understood. Research using pre-pandemic data from the U.S. has found that WFH is associated with some advantages such as greater job satisfaction but also some disadvantages including more work-to-family interference and job stress [4,5,6]. Studies focused on workers in other country contexts have found that WFH leads to improvements in job satisfaction, psychological attitudes, and work-life balance [7,8,9]. There is also evidence that WFH is valued by workers at up to 8 percent of a wage increase [3,10,11,12] and related literature has shown that increases in wages are associated with increased well-being [13,14,15,16,17,18].

The analysis in this study relies on the 2021 Well-Being Module of the American Time Use Survey, which features several questions related to the quality of life of respondents while performing three randomly selected activities from daily time diaries. After performing a principal component analysis to collapse the quality-of-life measures into a single latent variable measuring overall emotional well-being, we jointly estimate the association between WFH and overall emotional well-being scores during major daily activities in a seemingly unrelated regression framework, conditional on a wide variety of individual, household, and employment characteristics that can influence daily time use.

This study contributes to the literature in several ways. First, we use recently released nationally representative data from the U.S. to analyze the contemporaneous association between WFH and overall emotional well-being during commonly performed daily activities over a period of time when WFH was prevalent in the U.S. Second, our results are useful for informing cost-benefit analyses associated with WFH. For instance, greater emotional well-being while WFH may help to explain evidence that WFH boosts work productivity [7,8,9,19] and that workers who WFH typically earn higher wages [20,21]. Any boon to working productivity on home-based workdays operating through boosted emotional well-being is a potential benefit for employers. Finally, although access to remote work increased and became more diverse during the COVID-19 pandemic, it is still unevenly distributed across occupations and socioeconomic groups [1], suggesting that a more even distribution of access and uptake of remote work across U.S. workers may also promote a more equitable distribution of emotional well-being while working.

The remaining portion of this paper is organized as follows. Section 2 explains the data used in our study, presents descriptive statistics to motivate our empirical model, and describes the methods used to produce the results presented in Section 3. In Section 4, we offer a brief discussion of our results along with study limitations and avenues for future research. Finally, Section 5 provides concluding remarks.

## 2. Materials & Methods

### 2.1. Data

Our analysis draws on data from the ATUS (American Time Use Survey), which is conducted by the USCB (U.S. Census Bureau) for the BLS (U.S. Bureau of Labor Statistics). The ATUS has been administered yearly since 2003 to a randomly selected subset of households that completed their eighth and final month of CPS (Current Population Survey) interviews [22]. For each ATUS household, one individual who is at least 15 years old is interviewed by a USCB representative for detailed information about the activities s/he performed the day before the interview (i.e., the time diary day). Specifically, ATUS respondents were asked to identify their *primary activity* for each hour from 4 a.m. on the day before the interview to 4 a.m. on the interview day, where they were when they performed the activity, and who else was present when the activity was performed. If they were engaged in more than one activity at a given time, ATUS respondents were asked to identify the activity they considered to be the primary activity. The ATUS contains numerous activities, including dozens of four-digit activity codes (e.g., *working*, *taking classes*, *housework*, *shopping*, *eating and drinking*, etc.). In addition to a daily time diary, the ATUS also includes a wide variety of individual, household, and employment characteristics.

From 1 March 2021 to 31 December 2021, a WBM (Well-Being Module) was included in the ATUS and contains information on how respondents felt during three randomly selected activities—happy, tired, stressed, sad, and pained—as well as how meaningful each of those activities was to them. Activities randomly selected by the USCB for the ATUS-WBM were required to satisfy two criteria. They had to be (i) at least five minutes in duration and (ii) the following six-digit activity codes were not eligible: *sleeping* (0101xx), *grooming* (0102xx), *personal activities* (0104xx), *don’t know or can’t remember* (500106), and *refusal or none of your business* (500105).

### 2.2. Activity-Level Sample

We now provide a detailed overview of the three selection criteria that we applied to the 2021 ATUS-WBM data to arrive at our activity-level regression sample. First, given our particular interest in analyzing heterogeneity in overall emotional well-being during major daily time allocations by place of work, we begin by retaining only the 2101 ATUS-WBM respondents—out of a total of 6902 ATUS-WBM participants—who spent a nonzero amount of time engaged in a main job during their time diary day, either exclusively at home or exclusively somewhere outside the home. Although one may be concerned that work effort could be unusually high or low on the time diary day, it is important to note that ATUS interview dates are random concerning the time devoted to work or any other activity the day before the interview. Second, out of the 2101 individuals who worked during their time diary day, 1838 of them also have valid information on all six emotional well-being questions and the explanatory variables we use to specify our regression model (see Table 1). Finally, to avoid an ad-hoc selection of primary activities for our regression analysis, we implement a data-driven selection rule. Specifically, our analysis is limited to the respondent-activity cells or “episodes” that make up a minimum of 5 percent of all episodes. This selection rule results in the following top 5 four-digit codes: *Working* (0501)*Travel related to work* (1805)*Relaxing and leisure* (1203)*Food and drink preparation, presentation, and clean-up* (0202)*Eating and drinking* (1101)

We do not, however, include *Travel related to work (1805)* in our analysis because only 4 respondents who exclusively WFH also engaged in work-related travel during their diary day. Since the quantity and quality of food depend on whether the food was prepared inside or outside the home [23,24,25,26,27,28], we use the information on where the activity is performed to separately consider *eating and drinking at home* and *eating and drinking away from home* in the regression analysis.

Taken together, the remaining top 4 four-digit activity codes account for 56 percent of all episodes in our analytical sample. (Please see Appendix Table A1 for a full listing of all the individual activities (i.e., six-digit activity codes) within each of the four-digit activity codes used in our analysis.) Of the 1838 individuals we mentioned above, 1648 individuals spent time engaged in at least one activity within the top 4 four-digit activity codes during their time diary day. Among these 1648 individuals, 602 have information on 1 activity from a single four-digit code, 553 have information on 2 activities from two separate four-digit codes, and 144 have information on 3 activities from three separate four-digit codes. There are also 150 individuals with information on 2 activities within the same four-digit code, 181 with information on 2 activities within the same four-digit code plus another activity within a separate four-digit code, and 18 individuals with information on 3 activities within the same four-digit code. Therefore, our final activity-level regression sample amounts to 3037 respondent-activity observations or episodes: (1 × 602) + (2 × (553 + 150)) + (3 × (144 + 181 + 18)).

### 2.3. Individual-Level Sample

On an average day from March to December 2021, among the 1648 individuals in our regression sample who worked during their diary day, we find that about half of their daily time endowment (49.8 percent or 11.9 h) was spent on activities within the top 4 four-digit activity codes (Appendix Table A2). Also, we find that, over March–December 2021, 29 percent of respondents exclusively WAFH (worked away from home) and 71 percent of respondents exclusively WAFH during their diary day. 

To account for the complex survey design of the ATUS, obtain nationally representative estimates, and estimate accurate standard errors for individuals who worked a main job on an average day over March–December 2021, we apply ATUS-WBM sampling weights in all analyses. In particular, we use the ATUS-WBM final sampling weights when calculating individual-level descriptive statistics and the ATUS-WBM final activity sampling weights in the activity-level regression analysis. Since the CPS and ATUS have stratified and clustered sampling procedures, we employed the balanced repeated replication method using the final and replicate weights and a Fay coefficient of 0.5 to generate standard errors that are more precise than a method assuming a random sample.

In Table 1, we provide summary statistics for a wide variety of individual, household, and employment characteristics. There are several statistically significant differences in worker characteristics by WFH status. For instance, compared with individuals who WAFH, those who WFH are 23 pp (percentage points) more likely to have pursued education beyond a bachelor’s degree and 30 pp less likely to be paid on an hourly basis. Given these and other statistically significant differences in worker characteristics by WFH status, all explanatory variables are always controlled for in the regression analysis to reduce the risk that estimated WFH effects on emotional well-being are due to differences in the types of workers in our estimation sample.

### 2.4. Methods

Since the main goal of this study is to estimate the association between WFH and overall emotional well-being and many aspects of well-being cannot be measured by a single question, we conducted a PCA (principal component analysis). The PCA allows us to reduce dimensionality and collapse the various measures of how respondents felt during commonly performed activities into a single latent variable that is intended to measure the respondent’s overall emotional well-being. In the PCA, all emotional well-being measures are coded to move in the same direction—from the worst state to the best state—and are standardized to have a mean of 0 and SD (standard deviation) of 1. 

In our initial attempt to reduce dimensionality using all six measures, a scree plot suggested that the first two components should be retained since they both have eigenvalues above 1 (Appendix Figure A1). The first and second components explain 38 percent and 21 percent of the total variance in the six underlying emotional well-being measures, respectively. However, notably, the meaningfulness measure has a weak correlation with the first component as well as generally small correlations with the other emotional well-being measures (Appendix Table A3). Since we would like to retain only one component for the regression analysis, we dropped the meaningfulness measure, and repeated the PCA with the remaining five emotional well-being measures. 

As shown in Figure 1, using the remaining five well-being measures—happy, tired, stressed, sad, and pained—the explanatory power of the first component jumps by 6 percentage points relative to our initial PCA attempt. Now, the first component explains 44 percent of the total variance in the five underlying emotional well-being measures, which is over twice the amount of variation explained by the second component and over thrice the amount of variation explained by each of the other components. Differently put, the largest drop in total variance explained occurs when moving from component 1 to component 2 and the first component is the only component with an eigenvalue above 1. Therefore, we use only the first component in the regression analysis, which is, as expected, strongly and significantly correlated with each of the five emotional well-being measures (Appendix Table A4). Our PCA analysis findings are consistent with those of a previous study performing a PCA on the emotional well-being measures in the activity-level 2010 ATUS-WBM data [29].

Given that we are interested in estimating the association between WFH and overall emotional well-being during several major daily time allocations, we allow regression error terms to be correlated across well-being equations and specified regression equations of the following form in a seemingly unrelated regression framework:(1)$$EW_{ij}=\alpha_{0}+\alpha_{1}WFH_{i}+\alpha_{2}X_{i}^{’}+\alpha_{3}UR_{i}+\alpha_{4}RESIDENCE{’}_{i}+\;\alpha_{5}INTERVIEW_{i}^{’}+\alpha_{6}CUMWORK_{ij}+\alpha_{7}ACTIVITY_{ij}^{’}+\varepsilon_{ij},$$
where *EW* is the standardized principal component of five emotional well-being measures—happy, tired, stressed, sad, and pained—or the overall emotional well-being score for individual *i* during a major four-digit activity code *j*; α_0_ is the intercept term; WFH is a binary variable equal to 1 if individual *i* WFH during the diary day (and 0 otherwise); *X* is a vector collecting a variety of individual, household, and employment characteristics—age, age squared, gender, presence of a spouse or partner, presence of household children under age 6, presence of household children ages 6–11, race/ethnicity, education level, the logarithm of the hourly wage, hourly worker, full-time worker, and occupation dummy variables—as well as dummy variables capturing how well rested individual *i* felt yesterday and *i’s* feelings yesterday relative to a typical day; *UR* is the state-level unemployment rate where individual *i* resides during the month of the interview (to account for effects of state macroeconomy fluctuations on health status and well-being [30]); *RESIDENCE* is a vector consisting of a dummy variables for the state of residence of individual *i* (to absorb all time-invariant state characteristics, including permanent state-level differences in health environments) and for whether the residence of individual *i* is in a metropolitan area; *INTERVIEW* is a vector of interview-related factors for individual *i* (dummy variables for month of interview and day of the week) that may influence emotional well-being during daily time allocations; *CUMWORK* is the cumulative amount of work performed by *i* until activity *j* began; *ACTIVITY* is a vector of attributes specific to activity *j* (the time of the day it was performed, its duration, and whether individual *i* interacted with anyone during the activity); and $\varepsilon_{i}$ is an idiosyncratic error term. To allow for arbitrary correlation among observations within the same state, standard errors are clustered at the state level. Since three activities were selected at random for each respondent, the sample sizes vary across the dependent variables. Therefore, since the estimation subsamples are not all the same size, joint estimation of the regressions in a seemingly unrelated regression framework increases the precision regression estimates [31]).

Given repeated observations on the same respondents, one may wonder if a within-person estimator can be employed in our context. Unfortunately, there are only 15 respondents who WFH and WAFH during their diary day with both of those work activities being captured in the ATUS-WBM data. With a total of 39 activity-level observations, there is insufficient variation in WFH status to exploit in estimation. Moreover, the effect of WFH on well-being may be different for those who WFH and WAFH on the same day compared with those who WFH one day and WAFH another day. For example, a person who works in the office for 8 h and brings work home and works 2 h at night may feel differently while WFH than a person who exclusively WFH for 10 h in a single day.

## 3. Results

Table 2 presents the estimated associations between WFH and overall emotional well-being scores. Since the dependent variables have been standardized to have a mean of 0 and an SD of 1, these estimated WFH effects can be interpreted as a percent of an SD. To further help interpret the regression results, the average overall emotional well-being scores—as measured by the principal component—for each regression subsample are shown right above each of the regression estimates. Interestingly, the average score for working is negative (−0.14), which may reflect the general disutility associated with work. The other activities have positive average scores, with emotional well-being while eating being the highest.

We estimate that, on average, individuals who WFH have an overall emotional well-being score while working that is 15 percent of an SD higher than those who WAFH, which is statistically significant at the 10 percent level. Given the general disutility of supplying labor market work, this result suggests that those who WFH have a less negative affect while performing work duties than those who WAFH, indicating that home-based work environments may generally improve the emotional well-being of workers when they exert labor effort.

The patterns that emerge for the remaining major activities suggest that the place where activities are performed is important for understanding differences in overall emotional well-being by WFH status. We find small and statistically insignificant effects of WFH on overall emotional well-being scores during several home-based activities—relaxing and leisure (−0.09 SD), food and drink preparation (+0.05 SD), and eating at home (+0.07 SD) By contrast, we find that, on average, individuals who WFH have significantly higher emotional well-being scores—amounting to 63 percent of an SD—while eating away from home than do those who WAFH. This WFH estimate is almost 9 times as large as the WFH estimate associated with eating at home and a Wald test for the equality of the regression coefficients reveals that these are significantly different from each other (*p* = 0.001). In Wald tests for the equality of the regression coefficients, the WFH estimate associated with eating away from home is also significantly different from the WFH estimate associated with food preparation (*p* = 0.032), relaxing and leisure (*p* = 0.0002), and working (*p* = 0.004). This result suggests that, given the relatively higher portion of the day spent inside homes by those who WFH, the act of eating may be more enjoyable and thus generates a better affect when those who WFH separate from their remote workstations and leave their homes to eat.

## 4. Discussion

The remote workforce has ballooned in recent years. Previous studies have shown that there is substantial variation in daily time use by where work is performed, both before the highly disruptive COVID-19 pandemic began [21,32] and during the first year of the pandemic [33]. For example, WFH is associated with more time spent cooking and eating at home and less time spent eating away from home [32,33]. These eating-related time use patterns may facilitate healthier diets since food prepared away from home tends to be less nutritious and more caloric than food prepared at home [23,24,25,26,27,28]. 

Of course, WFH may also have effects on daily lives that extend beyond physical health, which can be measured by indicators of psychological well-being. Indeed, previous research based on random assignment of WFH at a Chinese company found that workers who WFH reported significantly higher positive attitude scores and lower negative attitude scores within a year [8]. We find consistent evidence in our analysis of nationally representative data on U.S. workers. WFH is contemporaneously associated with better emotional well-being scores while working, which suggests that on average home environments may produce more positive working experiences than office environments. Taken together with evidence from randomized controlled trials showing that happiness improves productivity [34] and that WFH boosts productivity [7,8,9,19], our findings may be taken to suggest one reason why workers who WFH typically earn higher wages: boosted emotional well-being while WFH may promote higher work productivity [20,21]. 

Our findings also suggest that WFH does not universally boost overall emotional well-being across commonly performed daily activities. Notably, while WFH is associated with better overall emotional well-being while working, WFH does not seem to provide a similar well-being boost during several common home-based activities, including relaxing, preparing food, and eating at home. By contrast, home-based workdays appear to generate higher overall emotional well-being when eating out, which is one of America’s favorite pastimes. Individuals who WFH may have more enjoyable experiences eating away from home during workdays than do those who WAFH because a relatively higher portion of their workday is spent indoors.

A major strength of this study is that it provides a new analysis of the association between WFH and emotional well-being measures—which are not commonly observed in survey data—over a period of time when WFH was prevalent in the U.S. These WFH estimates provide insight into how WFH shapes the quality of a life day and therefore throw some light on how WFH regularly may influence the quality of life. However, this study has a couple of limitations. First, our estimated WFH effects do not represent the causal effects of WFH on emotional well-being since WFH was not randomly assigned. Although we controlled for a wide variety of characteristics, we cannot rule out the possibility of reverse causality or that an unobserved third factor explains our estimated WFH effects, and therefore future research on how emotional well-being responds to random WFH assignment in the U.S. is warranted. Second, given that the emotional well-being measures used in this study were recorded during activities performed on a single day, our findings cannot speak to the relationship running from permanent WFH status and emotional well-being in the long run. This is an important future research endeavor given that WFH has become more widespread in the U.S. workforce since the onset of the COVID-19 pandemic and prior research has found evidence that higher subjective ratings of well-being are associated with better physical health and longevity [35].

## 5. Conclusions

The COVID-19 pandemic caused a surge in remote work in the U.S., with 1 in 2 U.S. workers reporting WFH in May 2020 [2]. Research using pre-pandemic data found that remote work is associated with greater job satisfaction and work-life balance, but also with more work-to-family interference and job stress [4,5,6]. Since the impact of remote work on U.S. workers’ quality of life is not yet well understood, this study used recent nationally representative U.S. data to analyze the relationship between remote work and workers’ emotional well-being during common daily activities. 

We find evidence that the associations between WFH and the overall emotional well-being of workers vary by activity. On average, compared with those who WAFH, the overall emotional well-being scores of those who WFH are significantly higher while working (+15 percent of an SD) and eating away from home (+63 percent of an SD). By contrast, the estimated associations between WFH and several home-based activities—relaxing and leisure, food preparation, and eating at home—are smaller and statistically insignificant. Taken together, our results suggest that for those who WFH, eating away from home on the days they WFH may be important for their emotional well-being.

## Figures and Tables

**Figure 1 ijerph-20-03616-f001:**
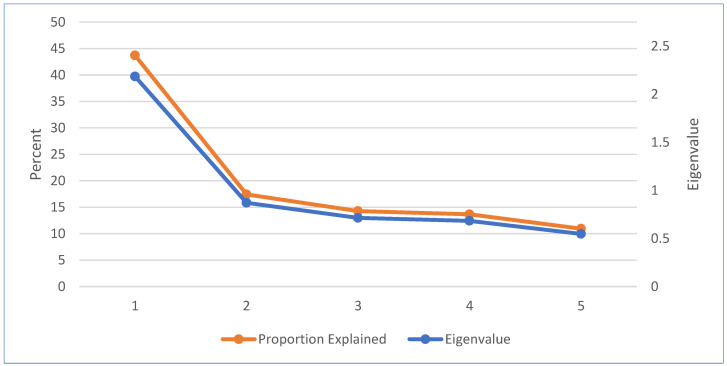
Percent of the total variance explained and eigenvalues from a principal component analysis using five well-being measures, by component. Notes: ATUS-WBM final activity sampling weights are used in this principal component analysis using five well-being measures (happy, tired, stressed, sad, and pained). N = 3037. For each component, the primary y-axis shows the percent of the total variance explained while the secondary y-axis shows the eigenvalue. Source: Authors’ calculations, using data from the BLS 2021 ATUS-WBM.

**Table 1 ijerph-20-03616-t001:** Descriptive statistics, 2021 ATUS-WBM, overall and by WFH status.

	Explanatory Variable		All	WFH	WAFH	Diff	
	**Age**	*years*	41.3	42.9	40.6	2.3	*
			(0.367)	(0.615)	(0.475)		
	**Male**	*percent*	57.0	51.2	59.4	−8.2	*
			(1.115)	(2.206)	(1.570)		
**Household**	**Spousal/partner present**	*percent*	60.7	66.5	58.3	8.2	*
			(1.399)	(2.421)	(1.788)		
	**Child/children under 18**	*percent*	32.5	35.4	31.4	4.0	
			(1.014)	(2.028)	(1.366)		
	**Child/children under 6**	*percent*	12.9	14.4	12.3	2.1	
			(0.815)	(1.552)	(1.067)		
	**Child/children between 6 and 11**	*percent*	15.8	15.0	16.1	−1.1	
			(0.875)	(1.446)	(1.144)		
	**Child/children between 12 and 17**	*percent*	15.8	14.8	16.2	−1.4	
			(0.917)	(1.594)	(1.123)		
**Ethnicity and race**	**Hispanic**	*percent*	20.1	9.2	24.5	−15.3	***
			(0.957)	(1.498)	(1.288)		
	**Non-Hispanic White**	*percent*	60.7	69.5	57.2	12.3	**
			(1.094)	(2.257)	(1.454)		
	**Non-Hispanic Black**	*percent*	11.4	9.9	12.0	−2.1	
			(0.733)	(1.485)	(0.982)		
	**Non-Hispanic Other**	*percent*	7.7	11.3	6.3	5.0	*
			(0.773)	(1.487)	(0.912)		
**Education level**	**Lower than high school**	*percent*	7.7	0.5	10.7	−10.2	***
			(0.775)	(0.418)	(1.096)		
	**High school degree or GED**	*percent*	25.6	6.1	33.5	−27.4	***
			(1.113)	(1.293)	(1.518)		
	**Some college or associate’s degree**	*percent*	22.9	15.3	26.0	−10.7	***
			(1.201)	(1.741)	(1.578)		
	**Bachelor’s degree**	*percent*	27.4	44.0	20.7	23.3	***
			(1.209)	(2.510)	(1.364)		
	**More than a bachelor’s degree**	*percent*	16.4	34.2	9.2	25.0	***
			(0.936)	(2.352)	(0.869)		
**Feeling vs. yesterday**	**Feel better**	*percent*	21.3	20.3	21.8	−1.5	
			(1.283)	(1.988)	(1.626)		
	**Feel the same**	*percent*	67.1	63.7	68.5	−4.8	
			(1.516)	(2.531)	(1.764)		
	**Feel worse**	*percent*	11.6	16.0	9.8	6.2	*
			(0.925)	(2.146)	(0.955)		
**Level of rest**	**Very rested**	*percent*	31.1	26.6	33.0	−6.4	
			(1.226)	(2.115)	(1.631)		
	**Somewhat rested**	*percent*	45.9	49.6	44.4	5.2	
			(1.410)	(2.506)	(1.728)		
	**A little rested**	*percent*	16.9	16.9	17.0	−0.1	
			(1.076)	(2.012)	(1.306)		
	**Not at all rested**	*percent*	6.0	6.9	5.7	1.2	
			(0.688)	(1.253)	(0.870)		
**Work status**	**Live in a metropolitan area**	*percent*	87.4	93.4	84.9	8.5	***
			(1.023)	(1.234)	(1.319)		
	**Hourly wage**	*dollars*	28.5	38.4	24.5	13.9	***
			(0.619)	(0.926)	(0.794)		
	**Hourly worker**	*percent*	35.9	14.3	44.7	−30.4	***
			(1.320)	(1.843)	(1.771)		
	**Full-time worker**	*percent*	86.2	92.6	83.6	9.0	***
			(0.942)	(1.188)	(1.232)		
**Occupation**	**Management**	*percent*	19.5	36.5	12.6	23.9	***
			(1.040)	(2.416)	(1.078)		
	**Professional**	*percent*	26.2	43.6	19.1	24.5	***
			(1.325)	(2.536)	(1.501)		
	**Service**	*percent*	14.3	2.1	19.3	−17.2	
			(1.036)	(0.583)	(1.418)		
	**Sales**	*percent*	6.4	4.9	7.1	−2.2	
			(0.791)	(1.087)	(1.061)		
	**Office**	*percent*	10.0	9.0	10.4	−1.4	
			(0.865)	(1.516)	(1.074)		
	**Farming**	*percent*	1.2	1.1	1.3	−0.2	***
			(0.275)	(0.540)	(0.324)		
	**Construction**	*percent*	4.4	0.0	6.2	−6.2	**
			(0.599)	0.000	(0.830)		
	**Installation**	*percent*	3.3	1.2	4.1	−2.9	***
			(0.497)	(0.457)	(0.675)		
	**Production**	*percent*	6.4	0.5	8.9	−8.4	***
			(0.827)	(0.325)	(1.177)		
	**Transportation**	*percent*	8.2	1.0	11.1	−10.1	***
			(0.936)	(0.587)	(1.283)		
	**Number of observations**		1648	563	1085		

Notes: ATUS-WBM final sampling weights are used to compute nationally representative estimates and appropriate standard errors (in parentheses) that account for the ATUS-WBM survey design. Source: Authors’ calculations, using data from the BLS 2021 ATUS-WBM. The symbols *, **, and *** denote statistical significance at the 10 percent, 5 percent, and 1 percent levels, respectively.

**Table 2 ijerph-20-03616-t002:** Coefficients and standard errors from seemingly unrelated regressions of overall emotional well-being during major activities on WFH status and other worker characteristics.

Dep Var = Standardized First Component	(1)		(2)	(3)	(4)	(5)	
	Working		Relaxing and Leisure	Food/Drink Prep	Eating and Drinking at Home	Eating and Drinking away from Home	
Dep Var Mean	−0.14		0.25	0.17	0.39	0.40	
WFH	0.145	*	−0.090	0.050	0.072	0.629	***
	(0.080)		(0.104)	(0.153)	(0.094)	(0.149)	
R^2^	0.44		0.36	0.55	0.40	0.62	
Respondent-Activity Observations	1011		823	374	542	287	

Notes: ATUS final activity sampling weights are used to compute nationally representative coefficient estimates and appropriate standard errors. Robust standard errors that are adjusted for clustering at the state level and that account for the survey design appear in parentheses below coefficient estimates. Please refer to Appendix Table A1 to see which activities are included in each column. All dependent variables are mutually exclusive. Covariates included but not shown: *age, age squared, gender, presence of a spouse or partner, presence of household children under age 6, presence of household children ages 6–11, race/ethnicity, education level, the logarithm of the hourly wage, hourly worker, full-time worker, occupation, how well rested the individual felt yesterday and his/her feelings yesterday relative to a typical day, state-level unemployment rate, state of residence, whether the residence is in a metropolitan area, interview month, interview day of the week, the time of the day the activity was performed, the duration of the activity, whether the individual interacted with anyone during the activity, and the cumulative amount of time spent working until the activity was performed*. The total number of respondent-activity observations is 3037. Source: Authors’ calculations, using data from the BLS 2021 ATUS-WBM. The symbols * and *** denote statistical significance at the 10 percent and 1 percent levels, respectively.

## Data Availability

This research uses information from public use data of the American Time Use Survey: https://www.bls.gov/tus/ (accessed on 11 November 2022). All analyses were conducted using the survey-related commands in the statistical software package Stata, version 17. The code is available upon request.

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
