# Peer review of "Working from Home and Emotional Well-Being during Major Daily Activities"

_ijerph, 2023, doi:10.3390/ijerph20043616_

Round 1

Reviewer 1 Report

Dear Authors,

The paper is based on relevant research and an important topic. The sample is good and the article potential is unquestionable. On the other hand, the work should be revised so that it conforms to the required academic format, rather than the current format, which resembles an executive report.

Major improvements

The abstract is too long and is based on the obtained results. The abstract should be changed to include the objective of the research, the applied methodology and an indication of the key results.

The number of keywords is too large and contains qualifiers that are too general (work, for example). Authors should perform a selection of keywords and determine 5 that will truly represent the essence of the work.

It is not typical for the Introduction to present research results. Therefore, the text starting in line 67 (We find evidence …) and ending in line 88 (…well-being while working.) should be eliminated from the Introduction. Instead of the above text, the authors should describe the objectives of the research, the research gap, the purpose of the research and list the research questions (up to 3).

In the Materials & Methods section, the subtitle "Sample" should be introduced and the characteristics of the sample should be presented within it.

Conclusion is too long. Parts of the conclusion should be used to form a separate section Discussion or those parts should be added to the framework of the section Results. In conclusion, key remarks should remain.

The authors should state the practical implications of the obtained results, as well as the limitations of the study.

Minor improvements

The way references are used in the text is not in accordance with the technical requirements. Authors should adapt the use of in-text sources to the technical guidance provided. Also, the way of presenting the used literature at the end of the text in the References section is not in accordance with the technical requirements of the journal and should be changed.

Row 29: "...fewer than 5 percent of paid full workdays were WFH just before..." An abbreviation was introduced into the text without prior explanation. This should be corrected in such a way that first the abbreviation is given, and then the whole term is put in parentheses. For example: "...fewer than 5 percent of paid full workdays were WFH (work from home) just before...". This should be applied to all abbreviations used in the text.

Rows 34-40: It is not clear why the two sentences are in parentheses. The text should be changed to eliminate the parentheses.

Rows 54-59: Again a sentence in parentheses. The text should be changed to eliminate the parentheses.

Sentence "The main goal of this study is to estimate the association between WFH and overall emotional well-being while engaged in one of the top 4 most performed activities over March-December 2021." it should not be part of the Methods section, but rather should be in the Introduction.

Kind regards

Reviewer 2 Report

This paper analyzes the contemporaneous association between working from home and overall emotional well-being during commonly performed daily activities in a seemingly unrelated regressions framework. Authors use data from the 2021 Well-Being Module of the American Time Use Survey. Results indicate that working from home has heterogenous relationships with the overall emotional well-being of workers during major daily activities

The research question is rather innovative, clear and well defined and the the paper is accurate. The discussion of results is sound and well conducted, and there is a clear description of the limitations and future research opportunities.

The paper does read fluidly, and it is written in Standard English.

For all those reasons, I think there is an overall benefit to publishing this work.

·          Does the introduction provide sufficient background and include all relevant references?

The introduction clearly presents the research question.

·         Is the research design appropriate?

The research on which the paper is based seems well designed. The analysis proposed follows well-known quantitative methods that give robust results.

·         Are the methods adequately described?

Methods are correctly described.

·         Are the results clearly presented?

Results are clearly presented.

·         Are the conclusions supported by the results?

The conclusions are drawn appropriately based on the data presented.

Review Comments to the Author

Dear Authors, I really appreciated reading your paper and I think it should be published.

Thank you!

Reviewer 3 Report

the study was conducted appropriately and the manuscript is well-written

No significant problems found in the document. 

Round 2

Reviewer 1 Report

Dear authors,

I appreciate the effort you put into correcting the initial manuscript. I also noticed that the English has been improved so that the article is much more fluent to read.

In its current form, the article is suitable for publication.

Kind regards